# Nicotinamide Mononucleotide and Coenzyme Q10 Protects Fibroblast Senescence Induced by Particulate Matter Preconditioned Mast Cells

**DOI:** 10.3390/ijms23147539

**Published:** 2022-07-07

**Authors:** Tsong-Min Chang, Ting-Ya Yang, Huey-Chun Huang

**Affiliations:** 1Department of Applied Cosmetology, Hungkuang University, Taichung 43302, Taiwan; ctm@sunrise.hk.edu.tw; 2Department of Medical Laboratory Science and Biotechnology, College of Medicine, China Medical University, Taichung 40402, Taiwan; s9724053@gmail.com

**Keywords:** human dermal fibroblast, PM, oxidative stress, NF-κB, senescence

## Abstract

Particulate matter (PM) pollutants impose a certain degree of destruction and toxicity to the skin. Mast cells in the skin dermis could be activated by PMs that diffuse across the blood vessel after being inhaled. Mast cell degranulation in the dermis provides a kind of inflammatory insult to local fibroblasts. In this study, we evaluated human dermal fibroblast responses to conditioned medium from KU812 cells primed with PM. We found that PM promoted the production of proinflammatory cytokines in mast cells and that the cell secretome induced reactive oxygen species and mitochondrial reactive oxygen species production in dermal fibroblasts. Nicotinamide mononucleotide or coenzyme Q10 alleviated the generation of excessive ROS and mitochondrial ROS induced by the conditioned medium from PM-activated KU812 cells. PM-conditioned medium treatment increased the NF-κB expression in dermal fibroblasts, whereas NMN or Q10 inhibited p65 upregulation by PM. The reduced sirtuin 1 (SIRT 1) and nuclear factor erythroid 2-related Factor 2 (Nrf2) expression induced by PM-conditioned medium was reversed by NMN or Q10 in HDFs. Moreover, NMN or Q10 attenuated the expression of senescent β-galactosidase induced by PM-conditioned KU812 cell medium. These findings suggest that NMN or Q10 ameliorates PM-induced inflammation by improving the cellular oxidative status, suppressing proinflammatory NF-κB, and promoting the levels of the antioxidant and anti-inflammatory regulators Nrf2 and SIRT1 in HDFs. The present observations help to understand the factors that affect HDFs in the dermal microenvironment and the therapeutic role of NMN and Q10 as suppressors of skin aging.

## 1. Introduction

Positive associations between air pollution particulate matter (PM) exposure and adverse health-related outcomes have been demonstrated [1,2]. These suspended PMs, including PM10 and PM2.5, which are classified according to aerodynamic diameter, are detrimental microns due to their ability to penetrate deeply into the lungs and enter the circulatory system, which could cause harmful effects, worsen diseases and even increase morbidity. Notably, epidemiological studies have found that long-term exposure to PM increases the risk of metabolic syndrome incidence [3,4]. Currently, more evidence is available on the effects of PM, which are correlated with not only the exacerbation of cardiovascular diseases and respiratory systemic inflammation but also with the progression of inflammatory skin diseases and allergic reactions [5,6].

The adult dermis is regarded as a postmitotic tissue settled by specialized, rarely proliferating cells that are expected to function more on damage repair and stress response. Fibroblasts are the most abundant cell type of the dermis and contribute most to the features of skin plasticity by providing new extracellular matrices [7]. In addition to their role with respect to structural support, fibroblasts are capable of responding to tissue damage signals, including the presence of inflammatory molecules and changes in mechanical properties and oxygen levels [8]. It appears that harmful stress affects long-lived fibroblasts more severely by the loss of cellular maintenance and repair mechanisms than highly proliferative keratinocytes that are frequently replaced. The skin dermis also houses long-lived mast cells (MCs), which represent up to 8% of the total number of cells within the dermis [9]. The important role of MCs in skin immunity is suggested by their strategic location and their complex interconnections with various types of skin cells. To exhibit their appropriate role as immune sentinels in response to the environment, communication must exist between MCs and fibroblasts within the dermal microenvironment [10]. Mast cells release several types of proinflammatory molecules that subsequently amplify and sustain dangerous alerts to fibroblasts [10,11]. Published studies have indicated that PM could enhance the FcεRI-mediated signaling and MC function [12,13,14,15]. In this study, we used PM-exposed mast cells to investigate the interactive effects between dermal neighborhood cells, which offered biological evidence that PM-induced MC activation contributes to premature human fibroblast senescence.

Nicotinamide mononucleotide (NMN), as the precursor of NAD^+^, is expected to provide significant preventive effects on various pathophysiological changes that occur during aging [16,17]. NMN possesses beneficial longevity properties by restoring the activity of NAD^+^-dependent deacetylase sirtuin 1 (SIRT1), which in turn improves mitochondrial function and prevents cellular senescence [18,19]. Previous studies have shown that SIRT1 protects fibroblasts from oxidative burden and DNA damage [20,21].

Coenzyme Q10 (CoQ10, ubiquinone-10) is an endogenous lipophilic quinone that can lower levels of inflammation due to its antioxidant activity [22,23]. The malfunction of mitochondria results in ROS formation and acceleration of cellular aging, whereas maintenance of adequate antioxidants appears to induce reparations for oxidative damage with important implications in the senescence process and cell death through interfering with inflammation [24,25]. CoQ10 and NADH are two components of the mitochondrial inner membrane. Notably, administration of NMN or CoQ10 has been reported to have remarkable therapeutic effects on age-related diseases [26,27,28]. This study was performed cellular and molecular examination of the effects of NMN and CoQ10 on inflamed HDFs induced by PM-mediated allergic hyperresponsiveness.

## 2. Results

### 2.1. Sublethal Level of Treatments on Human Dermal Fibroblasts (HDFs)

The conditioned supernatant from PM-treated KU812 cells is proposed to be more reflective of in vivo dermal microenvironment behavior, as it reduces the physical interaction of PM with HDF membranes. We initially employed the MTS assay to measure the cytotoxicity of PM-conditioned medium (PCM) from PM-exposed KU812 cells, NMN or Q10, on HDFs. HDFs were treated with PCM for 8 h followed by 1 μM NMN or 100 nM Q10 for 16 h. The concentrations of NMN and Q10 use were based on previously reported studies [29,30]. Cell viability was expressed as a percentage compared to the control, which was treated with a supernatant collected from DMSO-stimulated KU812 cells. The viability of HDFs incubated with PM conditional supernatant was decreased to 90% compared with that of the control (Figure 1). Both NMN and Q10 exhibited no apparent cytotoxicity. Replacing PCM with the indicated concentration of NMN- or Q10-containing medium slightly increased HDF viability. The cell viability among various treatments was more than 90%, revealing that all the applied treatments exhibited no significant cytotoxic effect on HDFs. The PM activated mast cells to release several proinflammatory molecules, including IL-1β, IL-6, and TNF-α (Table 1), which are implicated in inflammatory responses.

### 2.2. Inhibitory Effects of NMN and Q10 on the Generation of Oxidative Stress

Intracellular ROS production and mitochondrial ROS concentrations were estimated. We probed the levels of cellular ROS with the cell-permeant dye DCFH-DA. The quantified fluorescence intensity was dramatically increased in the HDFs treated with PCM, whereas the ROS levels were reduced in cells treated with NMN (Figure 2a). Enhanced mitochondrial ROS generation is a characteristic of senescent cells. Oxidative stress was quantified by using the MitoSOX probe, a fluorometric superoxide-detecting dye that specifically targets mitochondria. The levels of mitochondrial superoxide (mROS) in PCM-cultured HDFs were higher than those in mock control HDFs and were significantly reduced in response to NMN or Q10 (Figure 2b). These results demonstrated that NMN and Q10 exhibit antioxidant activity and can effectively neutralize ROS and mROS induced by PCM in HDFs.

### 2.3. NMN and Q10 Downregulated Inflammatory and Antioxidant-Related Pathways in HDFs

NF-κB is a key transcriptional regulator of inflammatory responses and plays an important role in the regulation of inflammation and in the development of cellular injuries [31]. We further investigated whether the inhibition of ROS stress by NMN and Q10 is mediated through the NF-κB pathway. PCM enhanced the level of the inflammatory marker NF-kB. NMN treatment significantly inhibited PCM activation of NF-κB expression. The conditioned medium-induced activation of NF-κB was reduced by Q10 treatment (Figure 3a). The results indicated that NMN and Q10 could counteract the effects of oxidative and proinflammatory signals caused by PCM treatment by acting against p65 production.

The expression of proteins related to the Nrf2 pathway was also investigated. As shown in Figure 3b, PCM treatment produced a considerable lowering of Nrf2 levels with respect to the control group. NMN treatment increased the Nrf2 level with respect to the control and counteracted the depressive effect exerted by PCM. Q10 also increased Nrf2, which efficiently counteracted PCM pretreatment (*p* < 0.05).

SIRT1 expression presented a trend similar to that of Nrf2 (Figure 3c). PCM treatment markedly decreased SIRT1 in HDFs. The application of NMN restored SIRT1 expression to levels similar to those in the control group. Adding Q10 after PCM pretreatment significantly increased SIRT1. Our results demonstrated the antioxidative mechanisms of NMN and Q10 in the stress induced by the PM-stimulated allergic secretome.

### 2.4. NMN and Q10 Attenuate PM-Conditioned Medium-Induced Senescence

SA-β-gal staining was used as a general marker of cellular senescence. The pretreatment of HDFs with PCM induced senescence. The expression and the percentage of cells positive for SA-β-Gal were higher in cultures of PCM than in the control. NMN or Q10 attenuated the stimulatory effects of PCM on senescence markers (Figure 4). In addition, the levels of NF-κB, a pivotal regulator of the senescence phenotype, were also higher in HDFs cultured with PCM. Collectively, these data confirmed that the PM-induced mast cell secretome is stressful, corresponding with an intensified ROS and senescence-associated secretory phenotype (SASP) to long-lived dermal fibroblasts.

## 3. Discussion

The skin dermal layer houses fibroblasts and immune cells, such as macrophages and lymphocytes, and high numbers of mature mast cells. Previous reports have shown the existence of microdomains within the dermis in which leukocytes formed stable perivascular sheaths. These resident mast cells act as the first line of defenders against invading insults and prime adaptive immunity [32]. The present work imitated the biological habitation of mast cells, which should be more reflective of the dermal microenvironment by matching its frontline defender role, concomitantly reducing the direct physical contact of PM with fibroblasts. The secretome of mast cells stimulated with PM via the paracrine route spreading to surrounding HDFs was established to investigate the relevant mechanisms [33]. Our results suggest that mast cells exposed to PM act as initiators or busters of immune injuries. PM- irritated mast cells cause fibroblast senescence by spreading ROS stress, which in turn worsens dermal aging.

Altered cellular oxidative status is able to promote stress-induced premature senescence, as documented by a dose-dependent increase in β-gal positivity. Following incubation with PCM, a significant increase in ROS levels was observed, indicating increased cellular oxidative status as a consequence of impaired mitochondrial functionality. Accumulated intracellular oxidative stress is critical in steering cellular biochemical pathways toward adaptive antioxidant defenses such as the Nrf2 response or cellular dysfunction. After exposure to NMN or Q10, we observed a significant decrease in mitochondrial superoxide production as well as a decline in the accumulation of cellular ROS. Similarly, in our presented experimental model, attenuated oxidative stress also significantly reduced the levels of β-gal formation.

Nrf2 activation is a major process of cellular defense against oxidative damage. This regulatory mechanism controls the expression of antioxidant response element (ARE)-related genes whose protein products are involved in the induction of the expression of antioxidant enzymes to neutralize electrophilic substances and high ROS insults. Therefore, diminished Nrf2 activity contributes to increased oxidative stress and the pathogenesis of various diseases [34]. Previous studies have confirmed crosstalk between Nrf2 and NF-κB. Nrf2 deficiency elevates the expression of NF-κB, leading to increased production of inflammatory factors [35]. Our results indicated that PM enhances mast cell activation and that these stimuli cause cytosolic ROS overproduction in fibroblasts, inducing mitochondrial oxidative levels, negatively regulating Nrf2 and sensitizing the NF-κB inflammatory pathway. Since SIRT1 counterregulates the inflammatory factor NF-κB p65, our results further showed that PM-activated MCs promoted NF-κB accumulation and SIRT1 inhibition. In contrast, both NMN and Q10 are able to reverse the blockade of Nrf2, positively regulate SIRT1 and downregulate the expression of p65. Supplementation with the NAD^+^ precursor NMN could trigger a sequence of events culminating in SIRT1 expression that also confers mitochondrial rejuvenation and antioxidative and anti-inflammatory capacities in HDFs [36,37]. Coenzyme Q10 is a molecule that acts as an essential cofactor of the electron transport chain as well as an endogenous antioxidant [38,39]. Q10 supplementation blocked cellular senescence that could be associated with changes in redox states and mitochondrial function. The use of Q10 significantly upregulated SIRT1 levels along with an increase in Nrf2 expression compared with the PM-treated group. Furthermore, Q10 administration markedly decreased p65 levels and significantly improved the senescence phenotype. Our data demonstrated that Q10 protects HDFs from PM via the induction of SIRT1 and Nrf2 expression.

## 4. Materials and Methods

The cell line identified as an immature human basophilic leukocyte KU812 cell line was purchased from the Bioresource Collection and Research Center, Taiwan. Human dermal fibroblasts and the associated culture medium were purchased from PromoCell GmbH (Sickingenstr, Heidelberg, Germany). CoQ10 and NMN were purchased from Sigma–Aldrich Chemical Co. (Merck KGaA, Darmstadt, Germany). ELISA kits for cytokine assays were purchased from eBioscience (San Diego, CA, USA). Primary antibodies against GAPDH were purchased from Santa Cruz Biotechnology Inc. (Dallas, TX, USA). Primary antibodies against NF-κB p65, SIRT 1 and Nrf2 were purchased from Cell Signaling Technology, Inc. (Danvers, MA, USA). The urban particulate matter and chemical reagents used in this study were purchased from Sigma–Aldrich Chemical Co.

### 4.1. Preparation of Conditioned Medium

PM master stock was prepared at a concentration of 10 mg/mL in dimethyl sulfoxide (DMSO). Prior to the preparation of culture treatments, the master stock was sonicated in a sonic bath for 1 h at 25 °C with regular vortexing to avoid any agglomeration of particles. After the addition of PM to KU812 medium for 24 h, KU812 cells were removed, and the supernatant collected was denoted as PM-conditioned medium (labeled PCM in the figures). HDF experiments were performed within 1 h to limit any variability in supernatant preparation.

### 4.2. Cell Culture

KU812 cells were grown in RPMI-1640 medium containing 10% heat-inactivated fetal calf serum and 1% antibiotics under standard cell culture conditions (37 °C, 5% CO_2_ in a humidified incubator). KU812 cells were incubated with 10 µg/mL PM2.5 for 24 h. Primary human dermal fibroblasts (HDFs) were grown in Fibroblast Growth Medium 2 (PromoCell GmbH, Germany) under standard cell culture conditions (37 °C, 5% CO_2_ in a humidified incubator). For exposure experiments, fibroblasts were exposed to a supernatant obtained from PM-primed KU812 cells for 8 h, and a fresh medium containing 100 nM Q10 or 1 μM NMN was subsequently replaced for 16 h.

### 4.3. Cell Proliferation Assay

The viability of HDFs was determined using a CellTiter 96* aqueous nonradioactive cell proliferation assay (MTS; Promega, Madison, WI, USA). In brief, 20 μL MTS solution was added to each well of a 96-well plate and incubated for 2 h in a humidified incubator at 37 °C containing 5% CO_2_. The formazan product was quantitated by measuring the absorbance at 490 nm with a microplate spectrophotometer (Thermo Fisher Scientific Inc., Waltham, MA, USA).

### 4.4. ROS Measurements

ROS were analyzed by flow cytometry using a 5 μg/mL final concentration of 2′,7′-dichlorodihydrofluorescein diacetate (DCFH-DA; Sigma Aldrich, St. Louis, MO, USA). Similarly, to measure the production of mitochondrial superoxide, cells were incubated in 5 μM MitoSOX Red (Invitrogen, Carlsbad, CA, USA) for 10 min. Next, the cells were washed in PBS and analyzed by a microplate spectrophotometer.

### 4.5. Senescence Assay

SA-β-gal staining was performed according to the manufacturer’s instructions (Abcam, Cambridge, UK.). Fifteen random images were taken per well at 100× magnification, and after randomization, positive and negative cells were counted in a blinded fashion by using ImageJ software.

### 4.6. Western Blot Analysis

Cells were lysed in RIPA buffer containing 5 μg/mL aprotinin, 100 μg/mL phenylmethylsulfonyl fluoride, 1 μg/mL pepstatin A, and 1 mM ethylenediaminetetraacetic acid (EDTA) at 4 °C for 20 min. Total lysates were quantified using a microBCA kit (Thermo Fisher Scientific, Waltham, MA, USA). Proteins (20 μg) were resolved by SDS-polyacrylamide gel electrophoresis and electrophoretically transferred to a polyvinylidene fluoride membrane. The membrane was blocked in 5% fat-free milk in PBST (PBS with 0.05% Tween-20), followed by overnight incubation with the following primary antibodies diluted in PBST: p65 Ab (1:1000) and pSTAT5a Ab (1:1000). The primary antibodies were removed, and the membranes were washed extensively in PBST. Subsequent incubation with horseradish peroxidase-conjugated secondary antibodies (1:20,000, Santa Cruz Biotech, Dallas, TX, USA) was performed at room temperature for 2 h. The membrane was again extensively washed in PBST to remove any excess secondary antibodies, and the blots were visualized using an enhanced chemiluminescence reagent (GE Healthcare, South Jakarta, Indonesia).

### 4.7. Statistical Analysis

Data were obtained from three independent experiments and are presented as the means ± standard error (SD). The results were statistically evaluated using one-way analysis of variance (ANOVA) using the Statistical Package for the Social Sciences (SPSS, version 21. IBM, Endicott, NY, USA.). The *p* value of the data was compared with the control and calculated by Student’s t-test (* *p <* 0.05 was considered significant, ** *p <* 0.01, *** *p <* 0.001).

## 5. Conclusions

Collectively, these findings reveal that paracrine factors of PM-stimulated mast cells induce bystander fibroblast senescence via inflammatory p65 activation and ROS induction. These molecular mechanisms of cellular senescence are predominantly mediated by SIRT1 and Nrf2 downregulation. NMN or Q10 ameliorated PM-induced inflammation by improving the cellular oxidative status, suppressing proinflammatory NF-κB, and promoting the levels of the antioxidant and anti-inflammatory regulators Nrf2 and SIRT1 in HDFs. The present observations could elucidate the factors that affect HDFs in the dermal microenvironment of PM-exposed mast cells and reposition the therapeutic roles of NMN and Q10 as suppressors of skin aging.

## Figures and Tables

**Figure 1 ijms-23-07539-f001:**
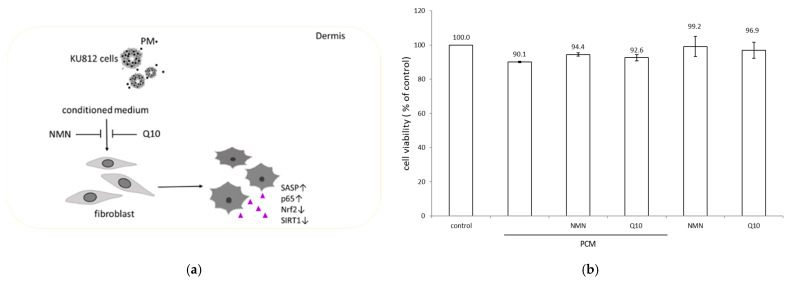
(**a**) Schematic depicting the experimental design. (**b**) Effect of PCM-conditioned medium, NMN and Q10 on HDF viability measured using the MTS Assay. All treatments were tested in triplicate, and the mean values ± SEM are shown.

**Figure 2 ijms-23-07539-f002:**
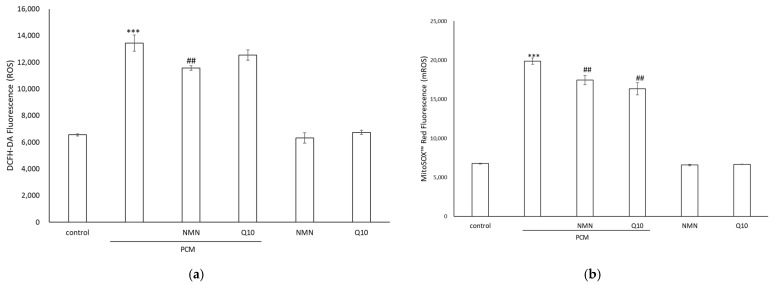
Representative levels of (**a**) intracellular ROS and (**b**) mitochondrial ROS (mROS) in HDFs treated with PCM or in combination with NMN or Q10. Data represent the mean ± SD of three independent experiments. *** *p* < 0.01 as compared with untreated control and *## p* < 0.01 as compared with PCM.

**Figure 3 ijms-23-07539-f003:**
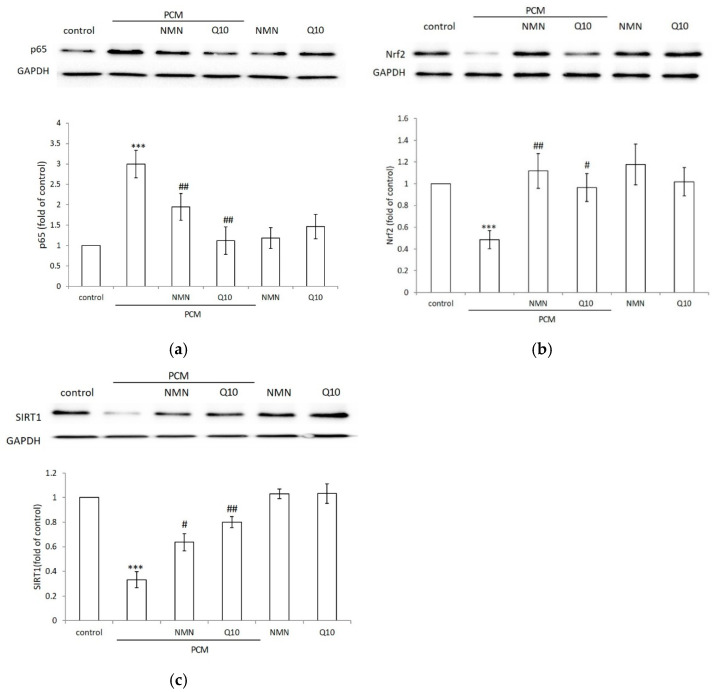
Representative western blots of (**a**) p-65, (**b**) Nrf2 and (**c**) SIRT1 in HDFs treated with mock control, PCM-conditioned medium, 1 μM NMN or 100 nM Q10 following incubation with PCM. The relative ratio of protein to total protein levels is presented as the mean ± SD (B). Fold change of control, *** *p* < 0.001. Fold change in PM-conditioned medium, # *p* < 0.05, ## *p* < 0.01.

**Figure 4 ijms-23-07539-f004:**
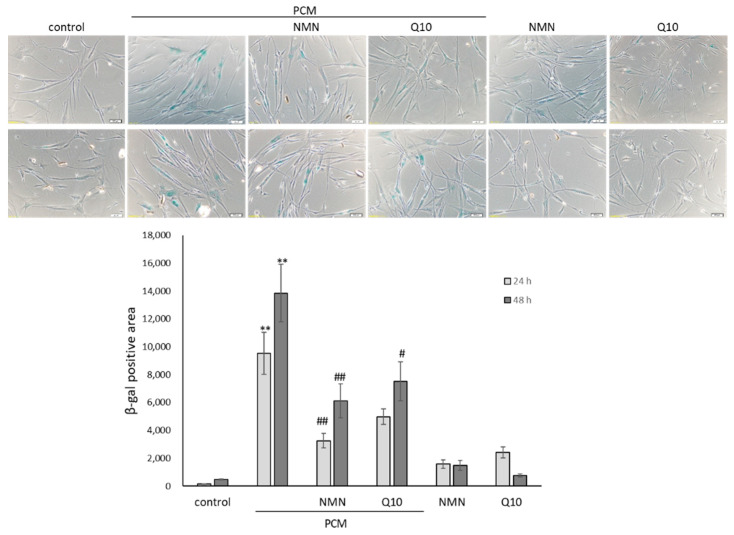
The effect of PCM on senescence in HDFs for 24 h and 48 h. The upper panel represents the cytochemical staining of SA-β-Gal (blue). The levels of positively stained area within various treatments were quantitatively compared in triplicate with mean values ± SEM. ** *p* < 0.01 compared with mock control and # *p* < 0.05, ## *p* < 0.01 compared with PM-conditioned medium group.

**Table 1 ijms-23-07539-t001:** Cytokine levels in the culture medium of PM-treated KU812 cells.

pg/mL	Control	PM
IL-6	15.01 ± 1.12	75.66 ± 1.03 ^b^
TNF-α	14.08 ± 0.56	26.02 ± 0.32 ^b^
IL-1β	145.10 ± 0.24	237.77 ± 11.73 ^a^

Compared to the control group, ^a^ *p* < 0.01, ^b^ *p* < 0.001.

## Data Availability

Not applicable.

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
