# Peer review of "Nicotinamide Mononucleotide and Coenzyme Q10 Protects Fibroblast Senescence Induced by Particulate Matter Preconditioned Mast Cells"

_ijms, 2022, doi:10.3390/ijms23147539_

Round 1

Reviewer 1 Report

In the present article the authors have demonstrated the beneficial effects of Q10 and NMN in fibroblasts exposed to PM exposed macrophage medium. This design simulates the effect of PM exposure in skin and could have important applications in skin aging induced by PM, skin inflammation and allergies. Indeed, the association of PMs with adverse health-related outcomes has been demonstrated.

The study is interesting and I have certain comments that need to be addressed.

1.      Why did the authors used 1 nM NMN or 100 nM CoQ10? The rational should be explained. Could increased doses offer additional positive effect?

2.      Line 19 “The reduced the silent information regulator, and the nuclear factor erythroid…” should probably be “The reduced silent information regulator 1, and nuclear factor erythroid 2…”

3.      Line 47 “…than by  highly proliferative keratinocytes…” by should be replaced by the

4.      Figure 1b should be enlarged and there is no clear description of the bars. The second bar is not labelled, there are two bars with NMN and two with Q10.

5.      I strongly recommend the authors to add the citation Sotiropoulou et al. 2021 Redirecting drug repositioning to discover innovative cosmeceuticals. Exp Dermatol 30: 628-644. This paper introduces the concept of repositioning (not only drugs but other compounds as well, natural products and compounds, chemical etc) for cosmetics. Q10 could be classical compound since it is used as supplement, is a natural occurring compound and is repositioning in this sense for the treatment of skin inflammation.

Author Response

Response to Reviewer                                 Jul 4, 2022

Manuscript ID: ijms-1793843

Title: Coenzyme Q10 Protects Fibroblast Senescence Induced by Particulate Matter Preconditioned Mast Cells

Authors: Tsong-Min Chang 1, Ting-Ya Yang 2, and Huey-Chun Huang 2*

Dear reviewers

Thank you for reviewing our manuscript entitled “Coenzyme Q10 Protects Fibroblast Senescence Induced by Particulate Matter Preconditioned Mast Cells”. Our revisions in response to the reviewers’ comments are addressed below in a point-by-point manner accordingly. Many thanks again for your valuable comments and suggestions. We are looking forward to your positive decision on our article.

Reviewer 2 Report

Comments to the Author

This manuscript is well-designed and well-explains the purpose, method, and results of the study. In addition, results strongly supports NMN and Coenzyme Q10 protect fibroblast senescence induced by particulate Matter preconditioned mast cells

There are some minor points which need to be considered for improving clarity of the mauscript.

1. The skin aging inhibitory effects of NMN and Q10 are mentioned throughout the study, but only Q10 is mentioned in the title. Is there any special reason?

2. Fig.1(b), the graph in the manuscript is incomplete.The X-axis item of the graph needs to be corrected.

3. Fig.3 (a), the p65 expression level of the Q10 treatment group not treated with PCM was higher than that of the control group. Supplementary explanation is required.

4.From Fig 4., it can be seen that the beta-gal positive area value of the NMN and Q10 treatment groups (without PCM treatment) was higher than that of the control group. A supplementary explanation is needed for this phenomenon.

Author Response

(The authors gave the same response as above.)
